# Learning MLPs on Graphs: A Unified View of Effectiveness, Robustness, and Efficiency

**Yijun Tian**[1], **Chuxu Zhang**[2], **Zhichun Guo**[1], **Xiangliang Zhang**[1], **Nitesh V. Chawla**[1]

[1]University of Notre Dame, [2]Brandeis University

`yijun.tian@nd.edu, chuxuzhang@brandeis.edu, zguo5@nd.edu,`
`xzhang33@nd.edu, nchawla@nd.edu`

## Abstract

While Graph Neural Networks (GNNs) have demonstrated their efficacy in dealing with non-Euclidean structural data, they are difficult to be deployed in real applications due to the scalability constraint imposed by the multi-hop data dependency. Existing methods attempt to address this scalability issue by training student multi-layer perceptrons (MLPs) exclusively on node content features using labels derived from the teacher GNNs. However, the trained MLPs are neither effective nor robust. In this paper, we ascribe the lack of effectiveness and robustness to three significant challenges: 1) the misalignment between content feature and label spaces, 2) the strict hard matching to teacher's output, and 3) the sensitivity to node feature noises. To address the challenges, we propose NOSMOG, a novel method to learn **NO**ise-robust **S**tructure-aware **M**LPs **O**n **G**raphs, with remarkable effectiveness, robustness, and efficiency. Specifically, we first address the misalignment by complementing node content with position features to capture the graph structural information. We then design an innovative representational similarity distillation strategy to inject soft node similarities into MLPs. Finally, we introduce adversarial feature augmentation to ensure stable learning against feature noises. Extensive experiments and theoretical analyses demonstrate the superiority of NOSMOG by comparing it to GNNs and the state-of-the-art method in both transductive and inductive settings across seven datasets. Codes are available at `https://github.com/meettyj/NOSMOG`.

## 1 Introduction

Graph Neural Networks (GNNs) have shown exceptional effectiveness in handling non-Euclidean structural data and have achieved state-of-the-art performance across a broad range of graph mining tasks (Hamilton et al., 2017; Kipf & Welling, 2017; Veličković et al., 2018). The success of modern GNNs relies on the usage of message passing architecture, which aggregates and learns node representations based on their (multi-hop) neighborhood (Wu et al., 2020; Zhou et al., 2020). However, message passing is time-consuming and computation-intensive, making it challenging to apply GNNs to real large-scale applications that are always constrained by latency and require the deployed model to infer fast (Zhang et al., 2020; 2022a). To meet the latency requirement, multi-layer perceptrons (MLPs) continue to be the first choice (Zhang et al., 2022b), despite the fact that they perform poorly in non-euclidean data and focus exclusively on the node content information.

Inspired by the performance advantage of GNNs and the latency advantage of MLPs, researchers have explored combining GNNs and MLPs together to enjoy the advantages of both (Zhang et al., 2022b; Zheng et al., 2022; Chen et al., 2021). To combine them, one effective approach is to use knowledge distillation (KD) (Hinton et al., 2015), where the learned knowledge is transferred from GNNs to MLPs through soft labels (Phuong & Lampert, 2019). Then only MLPs are deployed for inference, with node content features as input. In this way, MLPs can perform well by mimicking the output of GNNs without requiring explicit message passing, and thus obtaining a fast inference speed (Hu et al., 2021). Nevertheless, existing methods are neither effective nor robust, with three major drawbacks: (1) MLPs cannot fully align the input content feature to the label space, especially when node labels are correlated with the graph structure; (2) MLPs rely on the teacher's output to learn a

strict hard matching, jeopardizing the soft structural representational similarity among nodes; and (3) MLPs are sensitive to node feature noises that can easily destroy the performance. We thus ask: *Can we learn MLPs that are graph structure-aware in both the feature and representation spaces, insensitive to node feature noises, and have superior performance as well as fast inference speed?*

To address these issues and answer the question, we propose to learn **NO**ise-robust **S**tructure-aware **M**LPs **O**n **G**raphs (NOSMOG), a novel method with remarkable performance, outstanding robustness and exceptional inference speed. Specifically, we first extract node position features from the graph and combine them with node content features as the input of MLPs. Thus MLPs can fully capture the graph structure as well as the node positional information. Then, we design a novel representational similarity distillation strategy to transfer the node similarity information from GNNs to MLPs, so that MLPs can encode the structural node affinity and learn more effectively from GNNs through hidden layer representations. After that, we introduce the adversarial feature augmentation to make MLPs noise-resistant and further improve the performance. To fully evaluate our model, we conduct extensive experiments on 7 public benchmark datasets in both transductive and inductive settings. Experiments show that NOSMOG can outperform the state-of-the-art method and also the teacher GNNs, with robustness to noises and fast inference speed. In particular, NOSMOG improves GNNs by **2.05%**, MLPs by **25.22%**, and existing state-of-the-art method by **6.63%**, averaged across 7 datasets and 2 settings. In the meantime, NOSMOG achieves comparable efficiency to the state-of-the-art method and is **833×** faster than GNNs with the same number of layers. In addition, we provide theoretical analyses based on information theory and conduct consistency measurements between graph topology and model predictions to facilitate a better understanding of the model. To summarize, the contributions of this paper are as follows:

- We point out that existing works of learning MLPs on graphs are neither effective nor robust. We identify three issues that undermine their capability: the misalignment between content feature and label spaces, the strict hard matching to teacher's output, and the sensitivity to node feature noises.

- To address the issues, we propose to learn noise-robust and structure-aware MLPs on graphs, with remarkable effectiveness, robustness, and efficiency. The proposed model contains three key components: the incorporation of position features, representational similarity distillation, and adversarial feature augmentation.

- Extensive experiments demonstrate that NOSMOG can easily outperform GNNs and the state-of-the-art method. In addition, we present theoretical analyses, robustness investigations, efficiency comparisons, and ablation studies to validate the superiority of the proposed model.

## 2 RELATED WORK

**Graph Neural Networks.** Many graph neural networks (Veličković et al., 2018; Li et al., 2019; Zhang et al., 2019; Chen et al., 2020) have been proposed to encode the graph-structure data. They take advantage of the message passing paradigm by aggregating neighborhood information to learn node embeddings. For example, GCN (Kipf & Welling, 2017) introduces a layer-wise propagation rule to learn node features. GAT (Veličković et al., 2018) incorporates an attention mechanism to aggregate features. DeepGCNs (Li et al., 2019) and GCNII (Chen et al., 2020) utilize residual connections to aggregate neighbors from multi-hop and further address the over-smoothing problem. However, These message passing GNNs only leverage local graph structure and have been demonstrated to be no more powerful than the WL graph isomorphism test (Xu et al., 2019; Morris et al., 2019). Recent works propose to empower graph learning with positional encoding techniques such as Laplacian Eigenmap and DeepWalk (You et al., 2019; Wang et al., 2022; Tian et al., 2023a), so that the node's position within the broader context of the graph structure can be detected. Inspired by these studies, we incorporate position features to fully capture the graph structure and node positional information.

**Knowledge Distillation on Graph.** Knowledge Distillation (KD) has been applied widely in graph-based research and GNNs (Yang et al., 2020; Yan et al., 2020; Guo et al., 2023; Tian et al., 2023b). Previous works apply KD primarily to learn student GNNs with fewer parameters but perform as well as the teacher GNNs. However, time-consuming message passing is still required during the learning process. For example, LSP (Yang et al., 2020) and TinyGNN (Yan et al., 2020) introduce the local structure-preserving and peer-aware modules that rely heavily on message passing. To overcome the latency issues, recent works start focusing on learning MLP-based student models that do not require message passing (Hu et al., 2021; Zhang et al., 2022b; Zheng et al., 2022). Specifically, MLP student

is trained with node content features as input and soft labels from GNN teacher as targets. Although MLP can mimic GNN's prediction, the graph structural information is explicitly overlooked from the input, resulting in incomplete learning, and the student is highly susceptible to noises that may be present in the feature. We address these concerns in our work. In addition, since adversarial learning has shown great performance in handling feature noises and enhancing model learning capability (Jiang et al., 2020; Xie et al., 2020; Tian et al., 2022; Kong et al., 2022), we introduce the adversarial feature augmentation to ensure stable learning against noises.

## 3 PRELIMINARY

**Notations.** A graph is usually denoted as $\mathcal{G} = (\mathcal{V}, \mathcal{E}, \boldsymbol{C})$, where $\mathcal{V}$ represents the node set, $\mathcal{E}$ represents the edge set, $\boldsymbol{C} \in \mathbb{R}^{N \times d_c}$ stands for the $d_c$-dimensional node content attributes, and $N$ is the total number of nodes. In the node classification task, the model tries to predict the category probability for each node $v \in \mathcal{V}$, supervised by the ground truth node category $\boldsymbol{Y} \in \mathbb{R}^K$, where $K$ is the number of categories. We use superscript $^L$ to mark the properties of labeled nodes (i.e., $\mathcal{V}^L$, $\boldsymbol{C}^L$, and $\boldsymbol{Y}^L$), and superscript $^U$ to mark the properties of unlabeled nodes (i.e., $\mathcal{V}^U$, $\boldsymbol{C}^U$, and $\boldsymbol{Y}^U$).

**Graph Neural Networks.** For a given node $v \in \mathcal{V}$, GNNs aggregate the messages from node neighbors $\mathcal{N}(v)$ to learn node embedding $\boldsymbol{h}_v \in \mathbb{R}^{d_n}$ with dimension $d_n$. Specifically, the node embedding in $l$-th layer $\boldsymbol{h}_v^{(l)}$ is learned by first aggregating (AGG) the neighbor embeddings and then combining (COM) it with the embedding from the previous layer. The whole learning process can be denoted as: $\boldsymbol{h}_v^{(l)} = \text{COM}(\boldsymbol{h}_v^{(l-1)}, \text{AGG}(\{\boldsymbol{h}_u^{(l-1)} : u \in \mathcal{N}(v)\}))$.

## 4 PROPOSED MODEL

In this section, we present the details of NOSMOG. An overview of the proposed model is shown in Figure 1. We develop NOSMOG by first introducing the background of training MLPs with GNNs distillation, and then illustrating three key components in NOSMOG, i.e., the incorporation of position features (Figure 1 (b)), representational similarity distillation (Figure 1 (c)), and adversarial feature augmentation (Figure 1 (d)).

**Training MLPs with GNNs Distillation.** The key idea of training MLPs with the knowledge distilled from GNNs is simple. Given a cumbersome pre-trained GNN, the ground truth label $\boldsymbol{y}_v$ for any labeled node $v \in \mathcal{V}^L$, and the soft label $\boldsymbol{z}_v$ learned by the teacher GNN for any node $v \in \mathcal{V}$, the goal is to train a lightweight MLP using both ground truth labels and soft labels. The objective function can be formulated as:

$$\mathcal{L} = \sum_{v \in \mathcal{V}^L} \mathcal{L}_{GT}(\hat{\boldsymbol{y}}_v, \boldsymbol{y}_v) + \lambda \sum_{v \in \mathcal{V}} \mathcal{L}_{SL}(\hat{\boldsymbol{y}}_v, \boldsymbol{z}_v), \tag{1}$$

where $\mathcal{L}_{GT}$ is the cross-entropy loss between the student prediction $\hat{\boldsymbol{y}}_v$ and the ground truth label $\boldsymbol{y}_v$, $\mathcal{L}_{SL}$ is the KL-divergence loss between the student prediction $\hat{\boldsymbol{y}}_v$ and the soft labels $\boldsymbol{z}_v$, and $\lambda$ is a trade-off weight for balancing two losses.

**Incorporating Node Position Features.** To address the issue of misalignment between content feature and label spaces as well as assist MLP in capturing node positions on graph, we propose to enrich node content by positional encoding techniques such as DeepWalk (Perozzi et al., 2014). By simply concatenating the node content features with the learned position features, MLP is able to capture node positional information within a broad context of the graph structure and encode it in the feature space. The idea of incorporating position features is straightforward, yet as we will see, extremely effective. Specifically, we first learn position feature $\boldsymbol{P}_v$ for node $v \in \mathcal{V}$ by running DeepWalk algorithm on $\mathcal{G}$. Noticed that there are no node content features involved in this step, so the position features are solely determined by the graph structure and the node positions in the graph. Then, we concatenate (CONCAT) the content feature $\boldsymbol{C}_v$ and position feature $\boldsymbol{P}_v$ to form the final node feature $\boldsymbol{X}_v$. After that, we send the concatenated node feature into MLP to obtain the category prediction $\hat{\boldsymbol{y}}_v$. The entire process is formulated as:

$$\boldsymbol{X}_v = \text{CONCAT}(\boldsymbol{C}_v, \boldsymbol{P}_v), \qquad \hat{\boldsymbol{y}}_v = \text{MLP}(\boldsymbol{X}_v). \tag{2}$$

Later, $\hat{\boldsymbol{y}}_v$ is leveraged to calculate $\mathcal{L}_{GT}$ and $\mathcal{L}_{SL}$ (Equation 1).

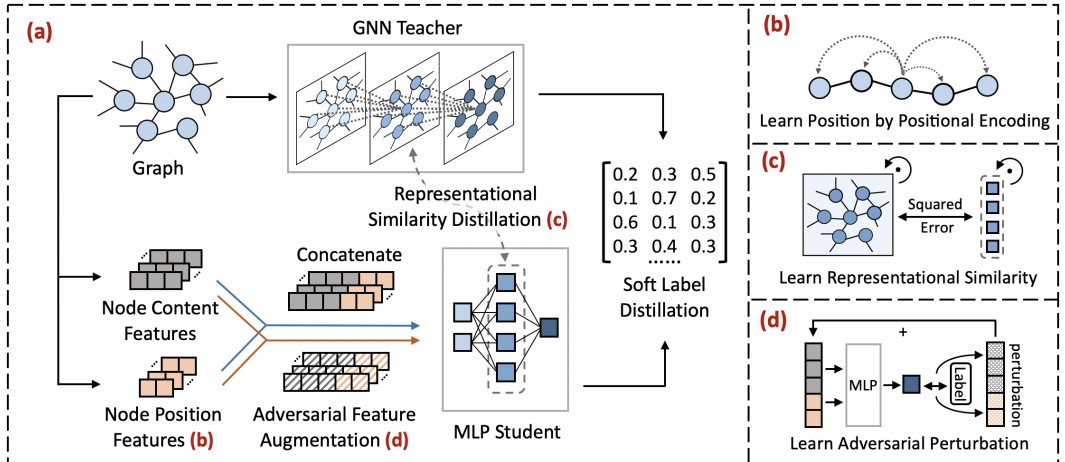

**Figure 1:** (a) The overall framework of NOSMOG: A GNN teacher is trained on the graph to obtain the representational node similarity and soft labels. Then, an MLP student is trained on node content features and position features, guided by the learned representational node similarity and soft labels. We also introduce the adversarial feature augmentation to ensure stable learning against feature noises. (b) Acquisition of Node Position Features: capturing node positional information by positional encoding techniques. (c) Representational Similarity Distillation: enforcing MLP to learn node similarity from GNN's representation space. (d) Adversarial Feature Augmentation: learning adversarial features by generating adversarial perturbation for input features.

**Representational Similarity Distillation.** With the aim of mitigating the constraint of strict hard matching to teacher's output, and facilitating MLP to capture the soft structural node similarity, we propose the Representational Similarity Distillation (RSD) to encourage MLP to learn from GNN's representation space. RSD preserves the similarity between node embeddings, and leverages mean squared error to measure the similarity discrepancy between GNN and MLP. Compared to soft labels, the relative similarity between intermediate representations of different nodes serves as a more flexible and appropriate guidance from the teacher model (Tung & Mori, 2019). Specifically, given the learned GNN representations $\boldsymbol{H}_G \in \mathbb{R}^{N \times d_G}$ and the MLP representations $\boldsymbol{H}_M \in \mathbb{R}^{N \times d_M}$, where $d_G$ and $d_M$ indicate the representation dimension of GNN and MLP respectively, we define the intra-model representational similarity $S_{GNN}, S_{MLP}$ for GNN and MLP as:

$$S_{GNN} = \boldsymbol{H}_G \cdot (\boldsymbol{H}_G)^T \qquad \text{and} \qquad S_{MLP} = \boldsymbol{H}'_M \cdot (\boldsymbol{H}'_M)^T, \quad \boldsymbol{H}'_M = \sigma(W_M \cdot \boldsymbol{H}_M), \quad (3)$$

where $W_M \in \mathbb{R}^{d_M \times d_M}$ is the transformation matrix, $\sigma$ is the activation function (we use ReLU in this work), and $\boldsymbol{H}'_M$ is the transformed MLP representations. We then define the RSD loss $\mathcal{L}_{RSD}$ to minimize the inter-model representation similarity using the Frobenius norm $|| \cdot ||_F$:

$$\mathcal{L}_{RSD}(S_{GNN}, S_{MLP}) = ||S_{GNN} - S_{MLP}||_F^2, \quad (4)$$

**Adversarial Feature Augmentation.** MLP is highly susceptible to feature noises (Dey et al., 2017) if only considers the explicit feature information associated with each node. To enhance MLP's robustness to noises, we introduce the adversarial feature augmentation to leverage the regularization power of adversarial features (Kong et al., 2022; Zhang et al., 2023). In other words, adversarial feature augmentation makes MLP invariant to small feature fluctuations and generalizes to out-of-distribution samples, with the ability to further boost performance (Wang et al., 2019). Compared to the vanilla training that original node content features $\boldsymbol{C}$ are utilized to obtain the category prediction, adversarial training learns the perturbation $\delta$ and sends maliciously perturbed features $\boldsymbol{X} + \delta$ as input for MLP to learn. This process can be formulated as the following min-max optimization problem:

$$\min_{\boldsymbol{\theta}} \left[ \max_{\|\delta\|_p \le \epsilon} \left( -\boldsymbol{Y} \log(\text{MLP}(\boldsymbol{X} + \delta)) \right) \right], \quad (5)$$

where $\boldsymbol{X}$ represents the concatenation of node content and position features as shown in Eq. 2, $\boldsymbol{\theta}$ indicates the model parameters, $\delta$ is the perturbation, $\| \cdot \|_p$ is the $\ell_p$-norm distance metric, and $\epsilon$ is the perturbation range. Specifically, we choose Projected Gradient Descent (Madry et al., 2018) as

the default attacker to generate adversarial perturbation iteratively:

$$\delta_{t+1} = \Pi_{\|\delta\|_\infty \leq \epsilon} \left[ \delta_t + \boldsymbol{s} \cdot \text{sign} \left( \nabla_\delta \left( -\boldsymbol{Y} \log(\text{MLP}(\boldsymbol{X} + \delta_t)) \right) \right) \right], \tag{6}$$

where $\boldsymbol{s}$ is the perturbation step size and $\nabla_\delta$ is the calculated gradient given $\delta$. For maximum robustness, the final perturbation $\delta = \delta_T$ is learned by updating Equation 6 for $T$ times to generate the worst-case noises. Finally, to accommodate KD, we perform adversarial training using both ground truth labels $\boldsymbol{y}_v$ for labeled nodes ($v \in \mathcal{V}^L$), and soft labels $\boldsymbol{z}_v$ for all nodes in graph ($v \in \mathcal{V}$). Therefore, we can reformulate the objective in Equation 5 as follows:

$$\boldsymbol{X}'_v = \boldsymbol{X}_v + \delta, \qquad \hat{\boldsymbol{y}}'_v = \text{MLP}(\boldsymbol{X}'_v),$$
$$\mathcal{L}_{ADV} = \max_{\delta \in \epsilon} [- \sum_{v \in \mathcal{V}^L} \boldsymbol{y}_v \log(\hat{\boldsymbol{y}}'_v) - \sum_{v \in \mathcal{V}} \boldsymbol{z}_v \log(\hat{\boldsymbol{y}}'_v)]. \tag{7}$$

The final objective function $\mathcal{L}$ is defined as the weighted combination of ground truth cross-entropy loss $\mathcal{L}_{GT}$, soft label distillation loss $\mathcal{L}_{SL}$, representational similarity distillation loss $\mathcal{L}_{RSD}$ and adversarial learning loss $\mathcal{L}_{ADV}$:

$$\mathcal{L} = \mathcal{L}_{GT} + \lambda \mathcal{L}_{SL} + \mu \mathcal{L}_{RSD} + \eta \mathcal{L}_{ADV}, \tag{8}$$

where $\lambda$, $\mu$ and $\eta$ are trade-off weights for balancing $\mathcal{L}_{SL}$, $\mathcal{L}_{RSD}$ and $\mathcal{L}_{ADV}$, respectively.

## 5 EXPERIMENTS

In this section, we conduct extensive experiments to validate the effectiveness, robustness, and efficiency of the proposed model and answer the following questions: 1) Can NOSMOG outperform GNNs, MLPs, and other GNNs-MLPs methods? 2) Can NOSMOG work well under both inductive and transductive settings? 3) Can NOSMOG work well with noisy features? 4) How does NOSMOG perform in terms of inference time? 5) How does NOSMOG perform with different model components? 6) How does NOSMOG perform with different teacher GNNs? 7) How can we explain the superior performance of NOSMOG?

### 5.1 EXPERIMENT SETTINGS

**Datasets.** We use five widely used public benchmark datasets (i.e., `Cora`, `Citeseer`, `Pubmed`, `A-computer`, and `A-photo`) (Zhang et al., 2022b; Yang et al., 2021), and two large OGB datasets (i.e., `Arxiv` and `Products`) (Hu et al., 2020) to evaluate the proposed model.

**Model Architectures.** For a fair comparison, we follow the paper (Zhang et al., 2022b) to use GraphSAGE (Hamilton et al., 2017) with GCN aggregation as the teacher model. However, we also show the impact of other teacher models including GCN (Kipf & Welling, 2017), GAT (Veličković et al., 2018) and APPNP (Klicpera et al., 2019) in Section 5.7.

**Evaluation Protocol.** For experiments, we report the mean and standard deviation of ten separate runs with different random seeds. We adopt accuracy to measure the model performance, use validation data to select the optimal model, and report the results on test data.

**Two Settings: Transductive vs. Inductive.** To fully evaluate the model, we conduct node classification in two settings: transductive (*tran*) and inductive (*ind*). For *tran*, we train models on $\mathcal{G}$, $\boldsymbol{X}^L$, and $\boldsymbol{Y}^L$, while evaluate them on $\boldsymbol{X}^U$ and $\boldsymbol{Y}^U$. We generate soft labels for every nodes in the graph (i.e., $\boldsymbol{z}_v$ for $v \in \mathcal{V}$). For *ind*, we randomly select out 20% test data for inductive evaluation. Specifically, we separate the unlabeled nodes $\mathcal{V}^U$ into two disjoint observed and inductive subsets (i.e., $\mathcal{V}^U = \mathcal{V}^U_{obs} \sqcup \mathcal{V}^U_{ind}$), which leads to three separate graphs $\mathcal{G} = \mathcal{G}^L \sqcup \mathcal{G}^U_{obs} \sqcup \mathcal{G}^U_{ind}$ with no shared nodes. The edges between $\mathcal{G}^L \sqcup \mathcal{G}^U_{obs}$ and $\mathcal{G}^U_{ind}$ are removed during training but are used during inference to transfer position features by average operator (Hamilton et al., 2017). Node features and labels are partitioned into three disjoint sets, i.e., $\boldsymbol{X} = \boldsymbol{X}^L \sqcup \boldsymbol{X}^U_{obs} \sqcup \boldsymbol{X}^U_{ind}$ and $\boldsymbol{Y} = \boldsymbol{Y}^L \sqcup \boldsymbol{Y}^U_{obs} \sqcup \boldsymbol{Y}^U_{ind}$. We generate soft labels for nodes in the labeled and observed subsets (i.e., $\boldsymbol{z}_v$ for $v \in \mathcal{V}^L \sqcup \mathcal{V}^U_{obs}$).

### 5.2 CAN NOSMOG OUTPERFORM GNNS, MLPS, AND OTHER GNNS-MLPS METHODS?

We compare NOSMOG to GNN, MLP, and the state-of-the-art method under the standard transductive setting and report the results in Table 1, which can be directly comparable to those reported in previous literature (Zhang et al., 2022b; Hu et al., 2020; Yang et al., 2021). As shown in Table 1,

**Table 1:** NOSMOG outperforms GNN, MLP, and the state-of-the-art method GLNN on all datasets under the standard setting. $\Delta_{GNN}$, $\Delta_{MLP}$, $\Delta_{GLNN}$ represents the difference between the NOSMOG and GNN, MLP, GLNN, respectively. Results show accuracy (higher is better).

| Datasets | SAGE | MLP | GLNN | NOSMOG | $\Delta_{GNN}$ | $\Delta_{MLP}$ | $\Delta_{GLNN}$ |
|---|---|---|---|---|---|---|---|
| Cora | $80.64 \pm 1.57$ | $59.18 \pm 1.60$ | $80.26 \pm 1.66$ | $\mathbf{83.04 \pm 1.26}$ | ↑ 2.98% | ↑ 40.32% | ↑ 3.46% |
| Citeseer | $70.49 \pm 1.53$ | $58.50 \pm 1.86$ | $71.22 \pm 1.50$ | $\mathbf{73.78 \pm 1.54}$ | ↑ 4.67% | ↑ 26.12% | ↑ 3.59% |
| Pubmed | $75.56 \pm 2.06$ | $68.39 \pm 3.09$ | $75.59 \pm 2.46$ | $\mathbf{77.34 \pm 2.36}$ | ↑ 2.36% | ↑ 13.09% | ↑ 2.32% |
| A-computer | $82.82 \pm 1.37$ | $67.62 \pm 2.21$ | $82.71 \pm 1.18$ | $\mathbf{84.04 \pm 1.01}$ | ↑ 1.47% | ↑ 24.28% | ↑ 1.61% |
| A-photo | $90.85 \pm 0.87$ | $77.29 \pm 1.79$ | $91.95 \pm 1.04$ | $\mathbf{93.36 \pm 0.69}$ | ↑ 2.76% | ↑ 20.79% | ↑ 1.53% |
| Arxiv | $70.73 \pm 0.35$ | $55.67 \pm 0.24$ | $63.75 \pm 0.48$ | $\mathbf{71.65 \pm 0.29}$ | ↑ 1.30% | ↑ 28.70% | ↑ 12.39% |
| Products | $77.17 \pm 0.32$ | $60.02 \pm 0.10$ | $63.71 \pm 0.31$ | $\mathbf{78.45 \pm 0.38}$ | ↑ 1.66% | ↑ 30.71% | ↑ 23.14% |

NOSMOG outperforms both teacher model and baseline methods on all datasets. Compared to the teacher GNN, NOSMOG improves the performance by 2.46% on average across different datasets, which demonstrates that NOSMOG can capture better structural information than GNN without explicit graph structure input. Compared to MLP, NOSMOG improves the performance by 26.29% on average across datasets, while the state-of-the-art method GLNN can only improve MLP by 18.55%, which shows the efficacy of KD and demonstrates that NOSMOG can capture additional information that GLNN cannot. Compared to GLNN, NOSMOG improves the performance by 6.86% on average across datasets. Specifically, GLNN performs poorly in large OGB datasets (the last 2 rows) while NOSMOG learns well and shows an improvement of 12.39% and 23.14%, respectively. This further demonstrates the effectiveness of NOSMOG. The analyses of the capability of each model component and the expressiveness of NOSMOG are shown in Sections 5.6 and 5.8, respectively.

## 5.3 CAN NOSMOG WORK WELL UNDER BOTH INDUCTIVE AND TRANSDUCTIVE SETTINGS?

To better understand the effectiveness of NOSMOG, we conduct experiments in a realistic production ($prod$) scenario that involves both inductive ($ind$) and transductive ($tran$) settings (see Table 2). We find that NOSMOG can achieve superior or comparable performance to the teacher model and baseline methods across all datasets and settings. Specifically, compared to GNN, NOSMOG achieves better performance in all datasets and settings, except for $ind$ on Arxiv and Products where NOSMOG can only achieve comparable performance. Considering these two datasets have a significant distribution shift between training data and test data (Zhang et al., 2022b), this is understandable that NOSMOG cannot outperform GNN without explicit graph structure input. However, compared to GLNN which can barely learn on these two datasets, NOSMOG improves the performance extensively, i.e., 18.72% and 11.01% on these two datasets, respectively. This demonstrates the capability of NOSMOG in capturing graph structural information on large-scale datasets, despite the significant distribution shift. In addition, NOSMOG outperforms MLP and GLNN by great margins in all datasets and settings, with an average of 24.15% and 6.39% improvement, respectively. Therefore, we conclude that NOSMOG can achieve exceptional performance in the production environment with both inductive and transductive settings.

## 5.4 CAN NOSMOG WORK WELL WITH NOISY FEATURES?

Considering that MLP and GLNN are sensitive to feature noises and may not perform well when the labels are uncorrelated with the node content, we further evaluate the performance of NOSMOG with regards to noise levels in Figure 2. Experiment results are averaged across various datasets. Specifically, we add different levels of Gaussian noises to content features by replacing $C$ with $\tilde{C} = (1 - \alpha)C + \alpha n$, where $n$ represents Gaussian noises that independent from $C$, and $\alpha \in [0, 1]$ incates the noise level. We find that NOSMOG achieves better or comparable performance to GNNs across different $\alpha$, which demonstrates the superior efficacy of NOSMOG, especially when GNNs can mitigate the impact of noises by leveraging information from neighbors and surrounding subgraphs, whereas NOSMOG only relies on content and position features. GLNN and MLP, however,

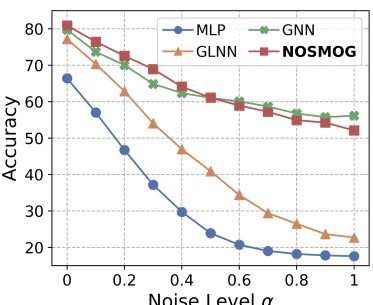

**Figure 2:** Accuracy vs. Feature Noises.

**Table 2:** NOSMOG outperforms GNN, MLP, and the state-of-the-art method GLNN in a production scenario with both **inductive** and **transductive** settings. *ind* indicates the results on $\mathcal{V}_{ind}^U$, *tran* indicates the results on $\mathcal{V}_{tran}^U$, and *prod* indicates the interpolated production results of both *ind* and *tran*.

| Datasets | Eval | SAGE | MLP | GLNN | NOSMOG | $\Delta_{GNN}$ | $\Delta_{MLP}$ | $\Delta_{GLNN}$ |
|---|---|---|---|---|---|---|---|---|
| Cora | *prod* | 79.53 | 59.18 | 77.82 | **81.02** | ↑1.87% | ↑36.90% | ↑4.11% |
| | *ind* | $81.03 \pm 1.71$ | $59.44 \pm 3.36$ | $73.21 \pm 1.50$ | $\mathbf{81.36 \pm 1.53}$ | ↑0.41% | ↑36.88% | ↑11.13% |
| | *tran* | $79.16 \pm 1.60$ | $59.12 \pm 1.49$ | $78.97 \pm 1.56$ | $\mathbf{80.93 \pm 1.65}$ | ↑2.24% | ↑36.89% | ↑2.48% |
| Citeseer | *prod* | 68.06 | 58.49 | 69.08 | **70.60** | ↑3.73% | ↑20.70% | ↑2.20% |
| | *ind* | $69.14 \pm 2.99$ | $59.31 \pm 4.56$ | $68.48 \pm 2.38$ | $\mathbf{70.30 \pm 2.30}$ | ↑1.68% | ↑18.53% | ↑2.66% |
| | *tran* | $67.79 \pm 2.80$ | $58.29 \pm 1.94$ | $69.23 \pm 2.39$ | $\mathbf{70.67 \pm 2.25}$ | ↑4.25% | ↑21.24% | ↑2.08% |
| Pubmed | *prod* | 74.77 | 68.39 | 74.67 | **75.82** | ↑1.40% | ↑10.86% | ↑1.54% |
| | *ind* | $75.07 \pm 2.89$ | $68.28 \pm 3.25$ | $74.52 \pm 2.95$ | $\mathbf{75.87 \pm 3.32}$ | ↑1.07% | ↑11.12% | ↑1.81% |
| | *tran* | $74.70 \pm 2.33$ | $68.42 \pm 3.06$ | $74.70 \pm 2.75$ | $\mathbf{75.80 \pm 3.06}$ | ↑1.47% | ↑10.79% | ↑1.47% |
| A-computer | *prod* | 82.73 | 67.62 | 82.10 | **83.85** | ↑1.35% | ↑24.00% | ↑2.13% |
| | *ind* | $82.83 \pm 1.51$ | $67.69 \pm 2.20$ | $80.27 \pm 2.11$ | $\mathbf{84.36 \pm 1.57}$ | ↑1.85% | ↑24.63% | ↑5.10% |
| | *tran* | $82.70 \pm 1.34$ | $67.60 \pm 2.23$ | $82.56 \pm 1.80$ | $\mathbf{83.72 \pm 1.44}$ | ↑1.23% | ↑23.85% | ↑1.41% |
| A-photo | *prod* | 90.45 | 77.29 | 91.34 | **92.47** | ↑2.23% | ↑19.64% | ↑1.24% |
| | *ind* | $90.56 \pm 1.47$ | $77.44 \pm 1.50$ | $89.50 \pm 1.12$ | $\mathbf{92.61 \pm 1.09}$ | ↑2.26% | ↑19.59% | ↑3.48% |
| | *tran* | $90.42 \pm 0.68$ | $77.25 \pm 1.90$ | $91.80 \pm 0.49$ | $\mathbf{92.44 \pm 0.51}$ | ↑2.23% | ↑19.66% | ↑0.70% |
| Arxiv | *prod* | 70.69 | 55.35 | 63.50 | **70.90** | ↑0.30% | ↑28.09% | ↑11.65% |
| | *ind* | $\mathbf{70.69 \pm 0.58}$ | $55.29 \pm 0.63$ | $59.04 \pm 0.46$ | $70.09 \pm 0.55$ | ↓-0.85% | ↑26.77% | ↑18.72% |
| | *tran* | $70.69 \pm 0.39$ | $55.36 \pm 0.34$ | $64.61 \pm 0.15$ | $\mathbf{71.10 \pm 0.34}$ | ↑0.58% | ↑28.43% | ↑10.05% |
| Products | *prod* | 76.93 | 60.02 | 63.47 | **77.33** | ↑0.52% | ↑28.84% | ↑21.84% |
| | *ind* | $\mathbf{77.23 \pm 0.24}$ | $60.02 \pm 0.09$ | $63.38 \pm 0.33$ | $77.02 \pm 0.19$ | ↓-0.27% | ↑28.32% | ↑11.01% |
| | *tran* | $76.86 \pm 0.27$ | $60.02 \pm 0.11$ | $63.49 \pm 0.31$ | $\mathbf{77.41 \pm 0.21}$ | ↑0.72% | ↑28.97% | ↑21.93% |

drop their performance quickly as $\alpha$ increases. In the extreme case when $\alpha$ equals 1, the input features are completely noises and $\tilde{C}$ and $C$ are independent. We observe that NOSMOG can still perform as good as GNNs by considering the position features, while GLNN and MLP perform poorly.

## 5.5 HOW DOES NOSMOG PERFORM IN TERMS OF INFERENCE TIME?

To demonstrate the efficiency of NOSMOG, we analyze the capacity of NOSMOG by visualizing the trade-off between prediction accuracy and model inference time on `Products` dataset in Figure 3. We find that NOSMOG can achieve high accuracy (78%) while maintaining a fast inference time (1.35ms). Specifically, compared to other models with similar inference time, NOSMOG performs significantly better, while GLNN and MLPs can only achieve 64% and 60% accuracy, respectively. For those models that have close or similar performance as NOSMOG, they need a considerable amount of time for inference, e.g., 2 layers GraphSAGE (SAGE-L2) needs 144.47ms and 3 layers GraphSAGE (SAGE-L3) needs 1125.43ms, which is not

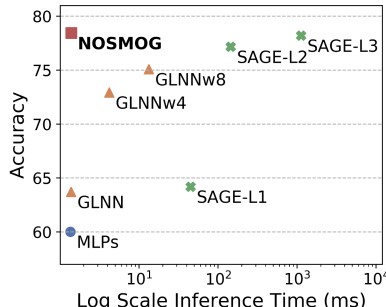

**Figure 3:** Accuracy vs. Inference Time.

applicable in real applications. This makes NOSMOG 107× faster than SAGE-L2 and 833× faster than SAGE-L3. In addition, since increasing the hidden size of GLNN may improve the performance, we compare NOSMOG with GLNNw4 (4-times wider than GLNN) and GLNNw8 (8-times wider than GLNN). Results show that although GLNNw4 and GLNNw8 can improve GLNN, they still perform worse than NOSMOG and even require more time for inference. We thus conclude that NOSMOG is superior to existing methods and GNNs in terms of both accuracy and inference time.

## 5.6 HOW DOES NOSMOG PERFORM WITH DIFFERENT MODEL COMPONENTS?

Since NOSMOG contains various essential components (i.e., node position features (POS), representational similarity distillation (RSD), and adversarial feature augmentation (ADV)), we conduct ablation studies to analyze the contributions of different components by removing each of them independently (see Table 3). From the table, we find that the performance drops when a component is removed, indicating the efficiency of each component. In general, the incorporation

**Table 3:** Accuracy of different model variants. The decreasing performance of these model variants demonstrates the effectiveness of each component in enhancing the model.

| Datasets | w/o POS | w/o RSD | w/o ADV | NOSMOG | $\Delta_{POS}$ | $\Delta_{RSD}$ | $\Delta_{ADV}$ |
|---|---|---|---|---|---|---|---|
| Cora | $80.44 \pm 1.51$ | $82.11 \pm 1.33$ | $82.43 \pm 1.42$ | $\mathbf{83.04 \pm 1.26}$ | ↑3.23% | ↑1.13% | ↑0.74% |
| Citeseer | $73.31 \pm 1.55$ | $71.61 \pm 1.84$ | $72.11 \pm 1.68$ | $\mathbf{73.78 \pm 1.54}$ | ↑0.64% | ↑3.03% | ↑2.32% |
| Pubmed | $75.55 \pm 2.54$ | $77.15 \pm 2.31$ | $77.02 \pm 2.58$ | $\mathbf{77.34 \pm 2.36}$ | ↑2.37% | ↑0.25% | ↑0.42% |
| A-computer | $82.94 \pm 0.87$ | $84.00 \pm 1.65$ | $83.15 \pm 1.21$ | $\mathbf{84.04 \pm 1.01}$ | ↑1.33% | ↑0.05% | ↑1.07% |
| A-photo | $92.76 \pm 0.58$ | $92.97 \pm 0.70$ | $92.24 \pm 0.98$ | $\mathbf{93.36 \pm 0.69}$ | ↑0.65% | ↑0.42% | ↑1.21% |
| Arxiv | $62.69 \pm 0.64$ | $71.59 \pm 0.27$ | $71.53 \pm 0.30$ | $\mathbf{71.65 \pm 0.29}$ | ↑14.29% | ↑0.08% | ↑0.17% |
| Products | $63.75 \pm 0.21$ | $78.38 \pm 0.36$ | $78.35 \pm 0.40$ | $\mathbf{78.45 \pm 0.38}$ | ↑23.06% | ↑0.09% | ↑0.13% |

of position features contributes the most, especially on `Arxiv` and `Products` datasets. By integrating the position features, NOSMOG can learn from the node positions and achieve exceptional performance. RSD contributes little to the overall performance across different datasets. This is because the goal of RSD is to distill more information from GNN to MLP, while MLP already learns well by mimicking GNN through soft labels. ADV contributes moderately across datasets, given that it mitigates overfitting and improves generalization. Finally, NOSMOG achieves the best performance on all datasets, demonstrating the effectiveness of the proposed model.

### 5.7 HOW DOES NOSMOG PERFORM WITH DIFFERENT TEACHER GNNs?

We use GraphSAGE to represent the teacher GNNs so far. However, different GNN architectures may have different performances across datasets, we thus study if NOSMOG can perform well with other GNNs. In Figure 4, we show average performance with different teacher GNNs (i.e., GCN, GAT, and APPNP) across the five benchmark datasets. From the figure, we conclude that the performance of all four teachers is comparable, and NOSMOG can always learn from different teachers and outperform them, albeit with slightly diminished performance when distilled from APPNP, indicating that APPNP provides the least benefit for student. This is due to the fact that the APPNP uses node features for prediction

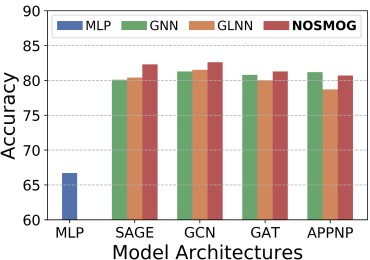

**Figure 4:** Accuracy vs. Teacher GNN Architectures.

prior to the message passing on the graph, which is very similar to what the student MLP does, and therefore provides MLP with little additional information than other teachers. However, NOSMOG consistently outperforms GLNN, which further demonstrates the effectiveness of the proposed model.

### 5.8 HOW CAN WE EXPLAIN THE SUPERIOR PERFORMANCE OF NOSMOG?

In this section, we analyze the superior performance and expressiveness of NOSMOG from several perspectives, including the comparison with GLNN and GNNs from information theoretical perspective, and the consistency measure of model predictions and graph topology based on Min-Cut.

**The expressiveness of NOSMOG compared to GLNN and GNNs.** The goal of node classification task is to fit a function $f$ on the rooted graph $\mathcal{G}^{[v]}$ with label $\boldsymbol{y}_v$ (a rooted graph $\mathcal{G}^{[v]}$ is the graph with one node $v$ in $\mathcal{G}^{[v]}$ designated as the root) (Chen et al., 2021). From the information theoretical perspective, learning $f$ by minimizing cross-entropy loss is equivalent to maximizing the mutual information (MI) (Qin et al., 2019), i.e., $I(\mathcal{G}^{[v]}; \boldsymbol{y}_i)$. If we consider $\mathcal{G}^{[v]}$ as a joint distribution of two random variables $\boldsymbol{X}^{[v]}$ and $\mathcal{E}^{[v]}$, that represent the node features and edges in $\mathcal{G}^{[v]}$ respectively, we have: $I(\mathcal{G}^{[v]}; \boldsymbol{y}_v) = I(\boldsymbol{X}^{[v]}, \mathcal{E}^{[v]}; \boldsymbol{y}_v) = I(\mathcal{E}^{[v]}; \boldsymbol{y}_v) + I(\boldsymbol{X}^{[v]}; \boldsymbol{y}_v | \mathcal{E}^{[v]})$, where $I(\mathcal{E}^{[v]}; \boldsymbol{y}_v)$ is the MI between edges and labels, which indicates the relevance between labels and graph structure, and $I(\boldsymbol{X}^{[v]}; \boldsymbol{y}_v | \mathcal{E}^{[v]})$ is the MI between features and labels given edges $\mathcal{E}^{[v]}$. To compare the effectiveness of NOSMOG, GLNN and GNNs, we start by analyzing the objective of GNNs. For a given node $v$, GNNs aim to learn an embedding function $f_{GNN}$ that computes the node embedding $\boldsymbol{z}_v$, where the objective is to maximize the likelihood of the conditional distribution $P(\boldsymbol{y}_v | \boldsymbol{z}^{[v]})$ to approximate $I(\mathcal{G}^{[v]}; \boldsymbol{y}_v)$. Generally, the embedding function $f_{GNN}$ takes the node features $X^{[v]}$ and its multi-hop neighbourhood subgraph $S^{[v]}$ as input, which can be written as $\boldsymbol{z}^{[v]} = f_{GNN}(X^{[v]}, S^{[v]})$.

Correspondingly, the process of maxmizing likelihood $P(\boldsymbol{y}_v|\boldsymbol{z}^{[v]})$ can be expressed as the process of minimizing the objective function $\mathcal{L}_1(f_{GNN}(X^{[v]}, S^{[v]}), \boldsymbol{y}_v)$. Since $S^{[v]}$ contains the multi-hop neighbours, optimizing $\mathcal{L}_1$ captures both node features and the surrounding structure information, which approximating $I(\boldsymbol{X}^{[v]}; \boldsymbol{y}_v|\mathcal{E}^{[v]})$ and $I(\mathcal{E}^{[v]}; \boldsymbol{y}_v)$, respectively.

GLNN leverages the objective functions described in Eq. 1, which approximates $I(\mathcal{G}^{[v]}; \boldsymbol{y}_v)$ by only maxmizing $I(\boldsymbol{X}^{[v]}; \boldsymbol{y}_v|\mathcal{E}^{[v]})$, while ignoring $I(\mathcal{E}^{[v]}; \boldsymbol{y}_v)$. However, there are situations that node labels are not strongly correlated to node features, or labels are mainly determined by the node positions or graph structure, e.g., node degrees (Lim et al., 2021; Zhu et al., 2021). In these cases, GLNN won't be able to fit. Alternatively, NOSMOG focuses on modeling both $I(\mathcal{E}^{[v]}; \boldsymbol{y}_v)$ and $I(\boldsymbol{X}^{[v]}; \boldsymbol{y}_v|\mathcal{E}^{[v]})$ by jointly considering node position features and content features. In particular, $I(\mathcal{E}^{[v]}; \boldsymbol{y}_v)$ is optimized by the objective functions $\mathcal{L}_2(f_{MLP}(X^{[v]}, P^{[v]}), \boldsymbol{y}_v)$ and $\mathcal{L}_3(f_{MLP}(X^{[v]}, P^{[v]}), f_{GNN}(X^{[v]}, S^{[v]}))$ by extending $\mathcal{L}_{GT}$ and $\mathcal{L}_{SL}$ in Equation 8 that incorporates position features. Here $f_{MLP}$ is the embedding function that NOSMOG learns given the content feature $X^{[v]}$ and position feature $P^{[v]}$. Essentially, optimizing $\mathcal{L}_3$ forces MLP to learn from GNN's output and eventually achieve comparable performance as GNNs. In the meanwhile, optimizing $\mathcal{L}_2$ allows MLP to capture node positions that may not be learned by GNNs, which is important if the label $\boldsymbol{y}_v$ is correlated to the node positional information. Therefore, there is no doubt that NOSMOG can perform better, considering that $\boldsymbol{y}_v$ is always correlated with $\mathcal{E}^{[v]}$ in graph data. Even in the extreme case that when $\boldsymbol{y}_v$ is uncorrelated with $I(\boldsymbol{X}^{[v]}; \boldsymbol{y}_v|\mathcal{E}^{[v]})$, NOSMOG can still achieve superior or comparable performance to GNNs, as demonstrated in Section 5.4.

**The consistency measure of model predictions and graph topology.** To further validate that NOSMOG is superior to GNNs, MLPs, and GLNN in encoding graph structural information, we design the cut value $\mathcal{CV} \in [0, 1]$ to measure the consistency between model predictions and graph topology (Zhang et al., 2022b), based on the approximation for the min-cut problem (Bianchi et al., 2019). The min-cut problem divides nodes $\mathcal{V}$ into $K$ disjoint subsets by removing the minimum number of edges. Correspondingly, the min-cut problem can be expressed

**Table 4:** The cut value. NOSMOG predictions are more consistent with the graph topology than GNN, MLP, and the state-of-the-art method GLNN.

| Datasets | SAGE | MLP | GLNN | NOSMOG |
|---|---|---|---|---|
| Cora | 0.9385 | 0.7203 | 0.8908 | **0.9480** |
| Citeseer | 0.9535 | 0.8107 | 0.9447 | **0.9659** |
| Pubmed | 0.9597 | 0.9062 | 0.9298 | **0.9641** |
| A-computer | 0.8951 | 0.6764 | 0.8579 | **0.9047** |
| A-photo | 0.9014 | 0.7099 | 0.9063 | **0.9084** |
| Arxiv | 0.9052 | 0.7252 | 0.8126 | **0.9066** |
| Products | 0.9400 | 0.7518 | 0.7657 | **0.9456** |
| Average | 0.9276 | 0.7572 | 0.8725 | **0.9348** |

as: $\max \frac{1}{K} \sum_{k=1}^{K} (\boldsymbol{C}_k^T \boldsymbol{A} \boldsymbol{C}_k)/(\boldsymbol{C}_k^T \boldsymbol{D} \boldsymbol{C}_k)$, where $\boldsymbol{C}$ is the node class assignment, $\boldsymbol{A}$ is the adjacency matrix, and $\boldsymbol{D}$ is the degree matrix. Therefore, we design the cut value as follows: $\mathcal{CV} = tr(\hat{\boldsymbol{Y}}^T \boldsymbol{A} \hat{\boldsymbol{Y}})/tr(\hat{\boldsymbol{Y}}^T \boldsymbol{D} \hat{\boldsymbol{Y}})$, where $\hat{\boldsymbol{Y}}$ is the model prediction output, and the cut value $\mathcal{CV}$ indicates the consistency between the model predictions and the graph topology. The bigger the value is, the predictions are more consistent with the graph topology, and the model is more capable of capturing graph structural information. The cut values for different models in transductive setting are shown in Table 4. We find that the average $\mathcal{CV}$ for NOSMOG is 0.9348, while the average $\mathcal{CV}$ for SAGE, MLP, and GLNN are 0.9276, 0.7572, and 0.8725, respectively. We conclude that NOSMOG achieves the highest cut value, demonstrating the superior expressiveness of NOSMOG in capturing graph topology compared to GNN, MLP, and GLNN.

## 6 CONCLUSION

In this paper, we address three significant issues of existing GNNs-MLPs frameworks and present a unified view of learning MLPs with effectiveness, robustness, and efficiency. Specifically, we propose to learn Noise-robust and Structure-aware MLPs on Graphs (NOSMOG) that considers position features, representational similarity distillation, and adversarial feature augmentation. Extensive experiments on seven datasets demonstrate that NOSMOG can improve GNNs by **2.05%**, MLPs by **25.22%**, and the state-of-the-art method by **6.63%**, meanwhile maintaining a remarkable robustness and a fast inference speed of 833× compared to GNNs. Furthermore, we present additional theoretical analyses as well as empirical investigations of robustness and efficiency to demonstrate the superiority of the proposed model.

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
