# OpenReview forum: "Learning MLPs on Graphs: A Unified View of Effectiveness, Robustness, and Efficiency"
_ICLR.cc/2023/Conference — ICLR 2023 notable top 25%_

### Official Review · Reviewer_8WF2 · 2022-10-22

**Confidence:** 4
**Correctness:** 3
**Technical Novelty And Significance:** 3
**Empirical Novelty And Significance:** 3
**Recommendation:** 8

**Clarity, Quality, Novelty And Reproducibility:**

The clarity and description of this work is relatively clear and understandable, and the novelty, the technique solutions, as well as the experiments denote the good quality of the work. Moreover, the work is not hard to reproduce.


**Strength And Weaknesses:**

strength

(1) this work explores the structure information and noise-robustness in knowledge distillation between GNNs and MLPs, which is a practical issue of this task for real-world applications. Specifically, structure information incorporation is a beneficial component for learning graph-structure data.

(2) the experimental parts are relatively comprehensive for illustrating the effectiveness of the proposed method. And some theoretical analysis, e.g., the information theory aspect, can support the claims of authors to some extent.


weakness:

(1) the solutions of this work for structure information capture with position encoding and noise robustness of adversarial training is straightforward, might limiting the novelty of this work.

(2) here are some questions and concerns required to be addressed:

a. in the inference stage of the trained MLP, is the node positional information also taken as the input together with the node content features? if so, compared with GLNN, additional information is used, so the performance improvement is a natural result. And whether the deepwalk time is counted in the inference time comparison in figure3?

b. it seems like the positional encoding features are only used in the L_GT and L_SL loss parts? are they also considered in the MLP representations HM for L_RSD loss and L_ADV loss？ if only in L_GT and L_SL, why do not utilize the positional encoding features for the remain components as the structure infomation of graphs is very important? if used for all loss terms, how to understand the performance in "w/o POS" in table 3 by removing each of them independently?

c. for the ablation study part, what about the ablation performance for only with the POS features? and what about the performance of NOSMOG without L_adv in figure 2? that would be a baseline result for verifying the performance of L_adv with nosiy features

d. except for feature dimensions, is there any other factors affecting the performance of POS? for example different encoding method? the scale of graphs? Consider the core of transfering knowledge from GNNs to MLPs is addressing the the scalability constraint under the multi-hop data dependency, would such structure information encoding also limit by the scalability?

e. for RSD part, why imposing extra transformation W in the the MLP representations HM? is this part kept in the inference? and according to my understanding, in eq.(3) S_gnn and S_mlp are both calculated with dot product? why it denotes the intra-model representational similarity ? what is the meaning of the intra-model similarity?


**Summary Of The Paper:**

This work deals with the knowledge distillation between the GNNs and MLPs from the perspectives of effectiveness and the robustness, and proposes a method NOSMOG to learn structure aware and noise robust MLP. For effectiveness, NOSMOG uses position encoding to capture graph structure information and combines it with original node content information for training MLP, along with the representation similarity distillation term. Moreover, adversarial training technique is introduced to address the robustness.


**Summary Of The Review:**

This work explores to the practical issues of structure infomation incorporation and noise-robustness in transfering knowledge from GNNs to MLP, and correpsonding solutions are fair reasonable and the effectiveness can be verified. however, there are still some parts to be further clarified and explained in the 'weakness' part above.

---

> ### Comment · Reviewer_8WF2 · 2022-11-17
> **Feedback**
>
> Thanks for the authors' response to my questions. My concerns were clarified, and I increased my rating.

---

> > ### Author Response · Authors · 2022-11-17
> > **Thank you**
> >
> > Dear Reviewer 8WF2,
> >
> > Thank you for increasing the score. We truly appreciate your effort in helping us to strengthen the paper and your support for our work!

---

### Official Review · Reviewer_44NK · 2022-10-24

**Confidence:** 4
**Correctness:** 4
**Technical Novelty And Significance:** 3
**Empirical Novelty And Significance:** 3
**Recommendation:** 8

**Clarity, Quality, Novelty And Reproducibility:**

The paper is clear and novel, and easily reproducible.


**Strength And Weaknesses:**

Strengths:
1. The problem is well-motivated, interesting, and important. The deployment of GNNs in real-world applications poses a fundamental scalability issue for the machine learning community. This paper addresses this scalability issue by training MLPs from GNNs, resolves three critical and pressing challenges that current methods have, and proposes a novel method to address the problems.

2. The authors present a not-complicated model with a strong capability to address the problems. Three key components are designed to learn an effective, robust, and efficient MLP. The proposed model can fully capture graph structure information in both the feature and representation spaces. In the meantime, the model is insensitive to node feature noises, with superior performance and fast inference speed. The proposed method is relatively novel to me.

3. The performance improvement is significant. The authors conduct extensive experiments and design thoughtful research questions to demonstrate the superiority of NOSMOG. For example, the proposed model can outperform the state-of-the-art by 12% to 23% on two large-scale OGB datasets.

4. The authors analyze the model expressiveness from 1) the information theoretical perspective and 2) the consistency measure of model predictions and graph topology based on Min-Cut. The authors explain well why the proposed model is better. The theoretical analyses are convincing.

5. The results and the claims are reproducible. The paper is well-written.


Weaknesses:

1. Descriptions on selecting the hyper-parameters can be introduced. In addition, considering the success of the proposed model relies on various hyper-parameters. Will the model be sensitive to different parameter values?

2. Additional background knowledge can be introduced for utilizing the min-cut problem to measure consistency.

3. Consistency measures for structure-only methods such as DeepWalk can be demonstrated for a comprehensive comparison. In particular, for Table 4, I am curious about the difference between the proposed method (that considers both content and structure) and DeepWalk which only considers the structural information.

**Summary Of The Paper:**

This paper proposes to learn MLPs on graphs to address the scalability issue while maintaining effectiveness, robustness, and efficiency. Specifically, the authors identify three significant challenges and propose a novel method NOSMOG. Furthermore, the authors provide theoretical analyses and conduct extensive experiments to demonstrate the superiority of the proposed model.

**Summary Of The Review:**

The results and findings in this paper are insightful and would be useful for future research. The paper is also well-written with solid theoretical exposition and strong results. Overall, this is a good paper.

---

### Official Review · Reviewer_iP1i · 2022-10-25

**Confidence:** 4
**Correctness:** 4
**Technical Novelty And Significance:** 4
**Empirical Novelty And Significance:** 3
**Recommendation:** 8

**Clarity, Quality, Novelty And Reproducibility:**

The paper is well-written and the approach is described clearly. The designed model is novel. The code and data are released for reproducibility.

**Strength And Weaknesses:**

Strengths:

+ The motivation for training student MLPs with teacher GNNs is interesting and important in practice. In addition, the authors identify and address three critical issues, which make the work applicable and significant in real-world cases. Really good motivation, especially since the graph datasets are becoming larger and scalability is becoming a pressing challenge.

+ The idea is compelling and the proposed model is somewhat novel. The authors present three components to address each identified problem correspondingly. The introduced components are simple yet effective. The overall model structure is well-designed. In the meantime, the model facilitates efficient inference, while achieving superior and robust performance.

+ This paper conducts extensive experiments and comprehensive ablation studies to evaluate the proposed model. For example, the authors compare both transductive and inductive settings, noisy input features, and different teacher architectures. The model is tested on multiple datasets and shows very promising results. The proposed model demonstrates state-of-the-art results across different settings and various scales of datasets. The results are remarkable and superior to other methods, which also suggests that NOSMOG can be very useful in practice from both performance and applicability perspectives. In addition, the authors open-source their implementation.

+ The authors provide a theoretical understanding of the model, which is sound. The theoretical statements in this paper look correct to me. The authors develop theoretical insights and build a theoretical basis for the proposed model, which I think is valuable.

Weaknesses and Questions:

- Some technical details are not elaborated for those readers who are less familiar with the respective topics. For example, the section on the consistency measure is quite brief. For better readability, I would suggest the authors provide more information such as how the cut value indicates and measures the consistency between model predictions and graph topology.

- Based on Table 3 in the ablation study, it seems that the POS contributes the most, especially on large graph datasets such as Arxiv and Products. I wonder what the performance looks like if only POS is used, instead of using all the components. Will the model variant have a similar or close performance to NOSMOG when it comes to large graph datasets?


**Summary Of The Paper:**

This work addresses three issues that current MLPs on graph methods have: 1) the misalignment between content feature and label spaces, 2) the strict hard matching to the teacher’s output, and 3) the sensitivity to node feature noises. In particular, this paper presents NOSMOG, a novel method to learn MLPs with remarkable effectiveness, robustness, and efficiency. Extensive experiments and theoretical analyses demonstrate the superiority of NOSMOG by comparing it to GNNs and state-of-the-art methods.

**Summary Of The Review:**

The paper address three challenges of learning MLPs on graphs by designing a novel and effective model. The technical statements are sound, the theoretical explanations are rigorous, and the empirical results are impressive.

---

### Official Review · Reviewer_yeLS · 2022-10-27

**Confidence:** 3
**Correctness:** 3
**Technical Novelty And Significance:** 2
**Empirical Novelty And Significance:** 3
**Recommendation:** 6

**Clarity, Quality, Novelty And Reproducibility:**

Clarity: The paper is clearly written and the components of NOSMOG are well described.
Quality: The experimental results are thorough. To improve the quality, it would be helpful to discuss the relationship between node position embeddings used at test time vs training time and to make section 5.8 more specific.
Novelty: Overall the paper provides incremental novelty building on the graph-less MLPs previously introduced.
Reproducibility: The authors provide code and detail the hyperparameters used.

**Strength And Weaknesses:**

Pros
* The authors are able to address limitations of current graph-less MLPs in a united system. Further the system (NOSMOG) is introduced clearly and each component is well explained.
* The experimental results are thorough and show the effectiveness of NOSMOG. The authors conduct evaluations in multiple settings (transductive vs inductive), run an ablation study, and test the robustness of using various teacher GNNs.

Cons
* Concatenating the node position features (embeddings) is a critical component of NOSMOG (contributes the most to an increase in accuracy), but the paper does not specifically discuss the process of re-embedding the graph during the testing phase and the impact of re-embedding the graph. For example, for the transductive setting, is the graph re-embedded at test time or are the embeddings for unlabeled nodes carried over from the training phase? Assuming re-embedding, it is possible that the re-embedded node positions are rotations or translations of the embeddings generated during training. It would be helpful to discuss whether NOSMOG is invariant to these operations on the node position embeddings during the test phase.
* Since many node position embeddings themselves are generated with GNNs today, it would be helpful to compare the performance of using non-message passing embeddings (i.e. the current DeepWalk embeddings) vs message passing embeddings. If NOSMOG improves with the message passing embeddings it would be helpful to discuss the tradeoff between latency and accuracy.
* Finally, the explanation of NOSMOG’s effectiveness in section 5.8 is quite abstract and I am not sure the discussion adds to the paper. As is the section largely repeats the justifications for incorporating the node position features in the first place. To improve the section it would be helpful to quantify the term $I(E^{[v]}; y_v)$ under certain modeling assumptions.

Minor comments
* Should one of the embedding matrices in equation 3 be transposed so that S is a gram matrix?
* I am assuming the MLP in equation 7 takes concatenated perturbed content features and node position embeddings as input though currently it only takes the content features.

**Summary Of The Paper:**

This work extends a recent approach to train MLPs with teacher GNNs to produce models capable of performing well on relational data while preserving low inference latency by omitting belief propagation. To motivate the paper, the authors identify two major limitations to existing MLPs trained with GNNs 1) loss of positional/structural information from the graph as the MLPs only take node content features as input 2) lack of robustness to node content perturbations.

To address these limitations the authors propose augmentations to the “graph-less” MLP training pipeline. The three augmentations are 1) concatenating node position embeddings (graph embeddings0 to the node content features for MLP input 2) representational similarity distillation such that the pairwise relationships between hidden representations of training nodes are preserved in the MLP compared to the GNN 3) training on adversarial samples for which the node content features are perturbed but the node labels remain untouched.

The authors provide extensive experimental results that show the performance of their augmented MLP on transductive and inductive node classification tests. The results show that the augmented graph-less MLP (named NOSMOG) outperforms the teacher GNN in most cases as well as the current SOTA graph-less MLP (GLNN). The ablation study shows that the first augmentation (node embeddings) most improves performance.

**Summary Of The Review:**

The paper presents a unified approach to address the limitations of current graph-less MLPs, a recent approach favored for faster inference latency, and overall extends the capabilities of knowledge distillation from GNNs to MLPs. The three augmentations are clearly explained and the experimental results are very thorough, showing the effectiveness of each component under various testing environments. The main limitation of the paper currently is greater insight into why the augmentations address the motivating limitations of graph-less GNNs and whether these augmentations themselves are robust. Specifically, the node position embeddings are the most important augmentation, as shown in the ablation study, but is unclear whether the MLP is invariant to symmetric operations on the node position embeddings when the graph is re-embedded at test time. Further, the theoretical explanation in section 5.8 would benefit from better quantifying the mutual information associated with the graph structure.

---

### Public Comment · ~Hui_Xu4 · 2022-11-17
**about the position embedding for test nodes in inductive setting**

For inductive setting, the position embeddings of test nodes are generated by aggregation, which leveraged the topology information during inference. However, glnn has not utilized topology information during inference. Whether is it a little unfair for glnn in your experiment?

---

> ### Author Response · Authors · 2022-11-17
> **Response to Hui Xu**
>
> Dear Hui,
>
> Thank you for your interest in our work and for pointing out your concern. As we demonstrate in the paper, we claim that GLNN does not well incorporate topology information, while we attend to address this problem. The usage of position features is one way of incorporating the topology information and we respectfully argue it is not an unfair comparison since this information is derived from the graph, which is accessible to GNNs, GLNN, and NOSMOG, in both the training and inference stages. In addition, we want to emphasize that the incorporation of topology information is one of our model designs, which is different from the previous method. To fully address your concern, we have conducted experiments to show the performance comparison between NOSMOG and GLNN with position features. The results are shown in the following table. From the table, we can see that NOSMOG can outperform GLNN with position features and achieves the best performance in all datasets, which again verifies the effectiveness of our model.
>
> | Methods | Cora | Citeseer | Pubmed | A-computer | A-photo | Arxiv | Products |
> | --- | --- | --- | --- | --- | --- | --- | --- |
> | GLNN + position features | 81.70 ± 1.31 | 71.43 ± 1.70 | 76.57 ± 2.89 | 83.50 ± 1.20 | 92.04 ± 0.76 | 71.27 ± 0.33 | 77.87 ± 0.34 |
> | NOSMOG | **83.04 ± 1.26** | **73.78 ± 1.54** | **77.34 ± 2.36** | **84.04 ± 1.01** | **93.36 ± 0.69** | **71.65 ± 0.29** | **78.45 ± 0.38** |
>
> In addition, we want to emphasize that in the inductive setting, the usage of topology information in the inference stage is a standard approach in the domain of graph learning [1, 2, 3]. Therefore, we follow these previous works to introduce the position features of the nodes in their immediate neighborhood. In particular, we leverage an average operator to combine the position features of the connected training nodes. Additional details about our transductive and inductive settings can be found in Section 5.1 of our paper. Furthermore, many GNN models consider the position features in the training and inference stages [4, 5, 6], which shows the validity of using position features.
>
> [1] Hamilton, Will, Zhitao Ying, and Jure Leskovec. "Inductive representation learning on large graphs." NeurIPS, 2017.
>
> [2] Zhang, Muhan, and Yixin Chen. "Inductive matrix completion based on graph neural networks." ICLR, 2020.
>
> [3] Chami, Ines, et al. "Hyperbolic graph convolutional neural networks." NeurIPS, 2019.
>
> [4] You, Jiaxuan, Rex Ying, and Jure Leskovec. "Position-aware graph neural networks." ICML, 2019.
>
> [5] Dwivedi, Vijay Prakash, et al. "Graph neural networks with learnable structural and positional representations." ICLR, 2022.
>
> [6] Li, Pan, et al. "Distance encoding: Design provably more powerful neural networks for graph representation learning." NeurIPS, 2020.

---

> > ### Public Comment · ~Hui_Xu4 · 2022-11-18
> > **FeedBack**
> >
> > Thanks for the authors' response to my questions.

---

### Decision · Program_Chairs · 2023-01-20

**Decision:**

Accept: notable-top-25%

**Justification For Why Not Higher Score:**

This is a good paper, but the method comes with a few caveats: it requires a pre-trained GNN and a run of DeepWalk to create positional embeddings, and fill require substantial hyperparameter tuning.

**Justification For Why Not Lower Score:**

This is a good paper that seems to suggest an extremely fast alternative to GNNs (although it requires a pre-trained GNN). It potentially opens the door to an avenue of new methods leveraging this concept to scale up GNN methods.

**Metareview: Summary, Strengths And Weaknesses:**

__Summary__ This paper proposes a new method to perform node classification on graphs that is considerably faster than GNNs, while allowing to outperform several GNN methods on established benchmarks. The crux of the proposed approach is to create graph positional embeddings (through deepwalk) and to append these embeddings to the node features before applying an MLP. The MLP is then trained using a teacher GNN (a pre-trained GNN model), so that the loss for training is a weighted combination of the cross entropy between the MLP predicted labels and the true ones, and the KL divergence between the MLP-predicted labels and that of the teacher GNN (a process inspired by Knowledge Distillation).  They also add a adversarial learning loss, as well as a representational similarity distillation loss to further encourage the learned MLP embeddings to resemble that of the GNN, and to alleviate the frailty of the MLP to adversarial attacks. The final loss is therefore the sum of 4 terms, with three hyperparameters to tune. The authors then perform an extensive suite of experiments to prove the methods competitiveness against GNNs in terms of running time and accuracy, and to show its superiority as a function of the feature noise level.

__Summary of the reviews.__ The reviews are unequivocally enthusiastic about this new approach. During the rebuttal, the authors addressed the few concerns raised by the reviewers, and provided further evaluation of their method --- in particular, using different representational embeddings than DeepWalk. The authors have also appropriately clarified a few unclear details in the main text.

**Note From Pc:**

if the above contains the word "oral" or "spotlight" please see: "oral" presentation means -> notable-top-5% and "spotlight" means -> notable-top-25%. As stated in our emails, we are disassociating presentation type from AC recommendations

**Summary Of Ac-Reviewer Meeting:**

N/A